# Bowel Perforation in Vascular Ehlers–Danlos Syndrome: Case Report and Comprehensive Review

**DOI:** 10.3390/jpm13081247

**Published:** 2023-08-10

**Authors:** Alexandra Menni, Georgios Tzikos, Alexandros Sarafis, Athina Ververi, George Chatziantoniou, Vasileios Rafailidis, Stavros Panidis, Patroklos Goulas, Eleni Karlafti, Stylianos Apostolidis, Olga Giouleme, Antonios Michalopoulos, Daniel Paramythiotis

**Affiliations:** 11st Propaedeutic Department of Surgery, AHEPA University Hospital, Aristotle’s University of Thessaloniki, 54636 Thessaloniki, Greece; giorgos-t@hotmail.com (G.T.); al_saras@hotmail.com (A.S.); gchantzian5@gmail.com (G.C.); st.panidis@gmail.com (S.P.); patroklos@live.com (P.G.); stlsa@auth.gr (S.A.); amichal@auth.gr (A.M.); danosprx1@hotmail.com (D.P.); 2Genetic Unit, 1st Department of Obstetrics & Gynaecology, Aristotle’s University of Thessaloniki, Papageorgiou University Hospital, 56429 Thessaloniki, Greece; athina.ververi@gosh.nhs.uk; 3Department of Radiology, AHEPA University Hospital, Aristotle’s University of Thessaloniki, 54636 Thessaloniki, Greece; biilraf@hotmail.com; 4Emergency Department, AHEPA University Hospital, Aristotle’s University of Thessaloniki, 54634 Thessaloniki, Greece; linakarlafti@hotmail.com; 52nd Propedeutic Department of Internal Medicine, Aristotle’s University of Thessaloniki, Hippokration University Hospital, 54642 Thessaloniki, Greece; olgagi@med.auth.gr

**Keywords:** vascular Ehlers–Danlos Syndrome, bowel perforation, case report, collagen, gene mutations

## Abstract

Introduction: Ehlers–Danlos syndromes (EDS) comprise a rare variety of genetic disorders, affecting all types of collagen. Herein, we describe a case of the vascular type of EDS, with coexisting segmental absence of intestinal musculature, while simultaneously performing a narrative review of the existing literature. Case Presentation: A 23-year-old male patient with a history of multiple abdominal operations due to recurrent bowel perforations and the presence of a high-output enterocutaneous fistula was admitted to our surgical department for further evaluation and treatment. After detailed diagnostic testing, the diagnosis of vascular-type EDS (vEDS) was made and a conservative therapeutic approach was adopted. In addition, a comprehensive review of the international literature was carried out by applying the appropriate search terms. Results: The diagnosis of vEDS was molecularly confirmed by means of genetic testing. The patient was treated conservatively, with parenteral nutrition and supportive methods. Thirty-four cases of bowel perforation in vEDS have been reported so far. Interestingly, this case is the second one ever to report co-existence of vEDS with Segmental Absence of Intestinal Musculature. Conclusions: Establishing the diagnosis of vEDS promptly is of vital significance in order to ensure that patients receive appropriate treatment. Due to initial non-specific clinical presentation, EDS should always be included in the differential diagnoses of young patients with unexplained perforations of the gastrointestinal tract.

## 1. Introduction

The Ehlers–Danlos syndromes (EDS) are an extremely rare genetic entity characterized by skin hyperelasticity, joint extensibility, and tissue fragility. They consist of 13 subcategories with the vascular-type one (vEDS) being extremely rare [1]. The suspicion of vEDS is usually raised as a consequence of its vascular complications, which are usually the initial manifestation, whereas some patients may present with the characteristic acrogeric phenotype [2]. The diagnosis of EDS is confirmed by genetic testing, which typically identifies a mutation in one of the genes expressing collagen proteins. Patients with vEDS have an increased risk of complications primarily affecting the gastrointestinal tract and the vascular system, which are potentially life-threatening. The disease onset is usually in childhood or adolescence and life expectancy is usually reduced. As concerns its management, due to lack of experience, there are no specific data for the best approach. In general, the patients’ are conservatively supported in terms of nutritional needs or protection from possible complications that may arise, which are usually treated surgically in an emergency situation, such as bowel rupture or acute bleeding. In this study, we present a rare case of a young male with vEDS, whose diagnosis escaped routine investigations for a long time. Furthermore, we have conducted a comprehensive search of the literature reported in Medline to identify all published cases of vEDS presenting intestinal perforation.

## 2. Case Presentation

A 23-year-old male was referred to the surgical department from another hospital for further investigation and management of a high-output enterocutaneous fistula and possible short bowel syndrome. The patient’s medical history included several operations in the abdominal cavity due to recurrent incidents of intestinal perforations. His surgical problems had begun two years earlier, when he was admitted to the emergency department with acute epigastric pain, migrating within the next few hours to the right iliac fossa. Laboratory tests revealed leukocytosis (white blood cells (WBC) count 15.32 k/μL, neutrophils 87%) and increased C-Reactive Protein (CRP) at 1.9 mg/dL (normal range: <0.5 mg/dL). An abdominal Computed Tomography scan (CT scan) revealed free intra-abdominal air and pneumatosis of the descending and sigmoid colon (Figure 1). Thus, an exploratory laparotomy was performed, revealing generalized fecal peritonitis and a rupture on the antimesenteric wall of the sigmoid colon and confirming the diagnosis of gastrointestinal (GI) tract perforation. The hole was primarily sutured and the patient was discharged from the hospital on the 9th postoperative day, experiencing an uncomplicated postoperative period.

Six months later, the patient presented again to the emergency unit with similar symptoms. His physical examination was indicative of acute abdomen pain. He had leukocytosis (WBC count: 14.8 k/μL, neutrophils: 89.4%). Again, the patient underwent an explanatory laparotomy which identified an extended area of inflammation of the ileum (20 cm proximal to the ileocecal valve). The mesentery appeared very fragile, and extensive bleeding occurred from the vessels of the affected area. Based on the intraoperative findings, the surgeons decided to perform a 35 cm long enterectomy of the terminal ileus and complementary appendectomy. The patient was discharged on the 10th postoperative day. Τhe pathology result reported lesions of acute serous enteritis of the small intestine, without other abnormal findings or evidence of malignancy.

However, four months later, the patient was presented to the emergency department of another tertiary hospital with acute abdominal pain. A new CT scan showed once again free intra-abdominal air, indicative of small bowel perforation. A third explanatory laparotomy took place. Intraoperatively, perforation of the ileus was observed and, therefore, a 30 cm long enterectomy of the small intestine was performed.

Once again, the biopsy of the resected specimen was not diagnostic. It identified dense inflammatory infiltrations of polymorphonuclear neutrophils mainly in the peripheral adipose tissue and serosa and to a lesser extent in the muscularis propria, submucosa, and mucosa. Findings were confirmed with immunohistochemical staining for SMA and desmin. In addition, mild inflammatory infiltrations of lymphocytes, plasma cells, and eosinophilic polymorphonuclear cells were identified in the mucous membrane; periodic acid–Schiff (PAS) and silver stains were negative for fungi, as was Congo’s stain. There was no evidence of idiopathic inflammatory bowel disease or malignancy.

The postoperative period was, this time, complicated. On the 9th postoperative day, the patient suffered from wound dehiscence. He was transferred to the operating room and, during the procedure, a new small bowel perforation was accidentally discovered and primarily sutured. For the closure of the abdomen, a dual-sided mesh was used. The patient remained hospitalized in order to receive nutritional support and undergo further follow-up. However, on the 34th postoperative day, he presented signs of mesh infection and underwent surgical mesh removal. During the operation, a small leak from the cecum was identified in the area of the appendectomy, which was considered to be the cause of the infection. Due to the site of the perforation being very close to the original anastomosis, cecectomy and an additional 30 cm long enterectomy of the terminal ileus were performed. Again, a new mesh was placed during the abdominal wall closure. For the first time, the pathology examination of the excised specimen identified multiple abnormal segments with a thinning wall and segmental absence of musculature. This was also confirmed by immunohistochemical stains of SMA (Smooth Muscle Actin) and desmin. Other sections of the bowel showed gradual mucosal ulceration, accompanied by edema of the intestinal wall, dilation and hyperemia of the vessels, and infiltrations by inflammatory cells, plasma cells, and polymorphonuclear cells.

Approximately one month later (65 days after his first admission), he developed a new surgical mesh infection and underwent an additional laparotomy. Intraoperatively, he was diagnosed with two new sites of small bowel perforation, which were primarily repaired. The infected mesh was removed and the abdomen was closed only by suturing the skin. Nine days postoperatively, he developed a high-output enterocutaneous fistula, which was treated conservatively with total parenteral nutrition, antibiotics, and octreotide.

Three months later, the patient was discharged from the hospital, with low output (about 10–25 cc/day) from the enterocutaneous fistula. One month post-discharge, he was re-admitted due to increase in the output (more than 500 cc per day). He underwent CT imaging, which showed a new small bowel rupture, so he underwent laparotomy for the 7th time. Intraoperatively, a longitudinal perforation of the small intestine and three more perforations of the small bowel were also identified. Therefore, a small-intestine enterectomy was performed and the sites of perforations were primarily restored.

The pathology report revealed epithelial ulceration and swelling of the mucosa, hemorrhagic seepage, and varying degrees of inflammatory infiltration of lymphocytes, plasma cells, and polymorphonuclear neutrophils throughout the wall; focally, severe thinning of the myelin sheath was observed, while no evidence of malignancy was documented. Approximately one week after the last surgery, the enterocutaneous fistula reappeared. The patient’s physical examination was normal and the fistula was treated conservatively.

Six months postoperatively, the patient was referred to our department for further diagnostic management and treatment of the remaining enterocutaneous fistula (Figure 2). A variety of diagnostic tests (hematological, endocrinological, gastrointestinal) and multiple imaging examinations were performed. There were no abnormal findings, except for vitamin Κ deficiency, as expected in the context of chronic non-intestinal feeding. The Magnetic Resonance Imaging (MRI) enterography confirmed the presence of a fistula between the jejunum and the skin, with the internal stoma of the fistula at a distance of about 65–70 cm from the Treitz (Figure 3). Therefore, given that the patient has a functional bowel slightly more than half a meter in length, the diagnosis of short bowel syndrome with the accompanying problems of malnutrition was established.

During his hospitalization, an attempted central venous catheter change was made (from right to left internal jugular vein) to continue his total parenteral nutrition. However, the X-ray performed after the placement of the catheter showed it was vertically placed in the middle of the left hemithorax and there was presence of fluid, which was indicative of hemothorax. The catheter was removed and the patient was transferred to the radiology department for a CT angiography. There was no active blood exsanguination at that time, and a chest tube for drainage was placed.

Based on the overall clinical presentation and the latest pathology report, which identified segmental loss of the bowel muscular layer, the suspicion of vEDS was raised and gene panel testing was performed. The latter identified a pathogenic variant c. 1662+1G>A in the COL3A1 gene, which is associated with vEDS. This sequence change affects a donor site in intron 23 of the COL3A1 gene and is expected to disrupt RNA splicing. Variants that disrupt the donor or acceptor splice site typically lead to a loss of protein function. The variant has been submitted multiple times to ClinVar and has been classified as pathogenic [3]. It has also been reported in affected individuals in the literature and has been shown to cause exon 23 skipping [4,5,6,7]. Parental testing was normal, confirming that the variant had likely arisen de novo in the patient. He subsequently underwent all appropriate investigations for individuals with vEDS and received genetic counseling [8] (Figure 4).

The patient’s hospitalization in our department lasted for a total of 82 days, during which he remained hemodynamically stable, experiencing an almost uncomplicated stay. The main aim was to enhance his nutrition due to the short bowel syndrome. He received a venous port catheter in order to continue the parenteral nutrition at home due to the persisting high-output enterocutaneous fistula (about 1300–2100 cc/day). In addition, the patient began to receive intestinal nutrition after contacting specialists on similar types of vEDS cases in the Saint Mark’s Hospital, UK, who provided input on how to improve the patient’s quality of life. Eventually, the patient was discharged from the hospital, receiving enteral nutrition and high-calorie parenteral nutrition at home, through the port catheter. At his last assessment, 2 months after discharge, the patient gained 6 kg, weighing 52.5 kg (body mass index: 16.4 kg/m^2^).

On the occasion of the case described above, a thorough review of the literature and an electronic literature search of PubMed databases was performed to detect all the published case reports pertinent to the presentation of any vEDS case. For literature search purposes, the subject heading “vascular Ehlers-Danlos”, “perforation”, and “bowel complications” combined with the MESH terms “case reports” with “and” as Boolean term were applied to retrieve data related to the objectives of this review. The literature search was focused on human studies and full-length articles (no abstracts) with no language restriction.

The titles and abstracts of all the studies identified were screened and assessed. Those that were obviously irrelevant were discarded. If eligibility could not be ascertained from the title or the abstract, the full text of the study was retrieved and the papers deemed suitable were reviewed for eligibility according to their clinical relevance. References in the selected papers were scrutinized for additional articles in a further effort to ensure that relevant publications were not missed. An ultimate check of databases was carried out on 28 January 2022.

## 3. Discussion

The Ehlers–Danlos Syndromes (EDS) are a rare group of inherited disorders of collagen synthesis. These syndromes are characterized by a wide range of clinical manifestations, as the genetic disorder in the regulation of collagen production affects various systems of the human body, such as the skin, the ligaments, the joints, the vessels through all over the body, and the intra-abdominal organs [9]. EDS was described at first by two dermatologists, Dane Edvard Ehlers in 1901 and the French Henri-Alexandre Danlos in 1908, both of whom described, independently, the correlation of joints’ hypermobility with recurrent spontaneous hematomas and fragile skin [10,11]. The EDS are inherited in an autosomal-dominant way, although cases with no clear heredity have been described [12,13]. The most serious clinical manifestations, which can be life-threatening, involve complications from the vascular or GI system requiring immediate intervention [14]. The EDS are clinically classified into three main categories: the Classical type (I, II), the Hypermobility type (III), and the Vascular type (IV) [1]. However, the “Villefranche” classification at 1998 identifies six sub-categories of the EDS according to the respective major and minor criteria. Thus, in addition to the three main groups mentioned above (I–II, III, IV), the Kyphoscoliotic type (VI), the Arthrochalasia type (VIIa+b), and the Dermatosparaxis (VIIc) type are also identified but remain extremely rare [15]. In 2017, a new classification of the syndrome was created. This classification was primarily based on the clinical presentation of the patients rather than their genetic profile. It includes 13 categories while maintaining the nominal classification of the previous classification since there is also a definite genetic differentiation [16].

Collagen is the main connective tissue in the human body and plays a primary role in the pathogenesis of the EDS. All EDS subtypes result from gene mutations that affect either the collagen itself or the modulating enzymes of its fibers, leading to reduced structural integrity [17]. A total of 28 types of collagens have been discovered, each of them presenting in specific locations among tissues. The most commonly identified collagen types are type I, which is located in a variety of tissues; type II and XI, found primarily in the cartilage; type III collagen, located in the skin, the blood vessels, the uterus, and the internal organs; and, finally, type V, which is produced together with type I [18]. Collagen types III and V are the main types that are eliminated or nonfunctioning in patients with EDS and are responsible for a wide variety of clinical manifestations [19].

In classical EDS, the major mutations involve two genes responsible for the synthesis of collagen type V: COL5A1 and COL5A2. However, in the vascular type, the responsible mutation is found in the COL3A1 gene, which transcribes type III collagen [9]. Type III collagen is the second most prevalent type of collagen, associated with type I in all soft tissues, especially in the skin, the blood vessels, and the GI tract, where it is responsible for up to 30% of the collagen content [20]. Like all types of collagens, it is a macro protein above 300 nm in length with a molecular weight of more than 300 kDa [21]. It promotes the structural support of tissues while influencing cellular behavior at surface receptors, playing an important role in wound healing, angiogenesis, and the process of clotting and hemostasis, as well as in cell growth and differentiation [22].

The overall incidence of the EDS is 1:1000 to 1:25,000, with the three main groups (I–IV) appearing more frequently [23]. Regarding vEDS, it accounts for less than 4% of all EDS [12]. Its prevalence is estimated to be approximately 1:20,000 to 1:50,000 and it is inherited in an autosomal-dominant way. The vEDS is possibly considered the most severe type of EDS with a median life expectancy of around 50 years [24]. The vEDS is caused by heterozygous mutations mainly in the COL3A1 gene, with more than 500 mutations having been identified until now. Approximately 65% of them substitute glycine residues in the canonical triplet repeats and about 25% consist of splice-site variants that result in the in-frame exon skipping [25]. All COL3A1 mutations lead to the creation of an early, premature final codon with subsequent mRNA instability. Regarding the relationship between genotype and phenotype of the patients, different types of gene mutation have different phenotypes in patients, determining the prognosis. In particular, null mutations are associated with the delayed onset of symptoms, over twenty years, with complications being limited to vascular events. Patients with in-frame exon-skipping splice variants as well as those with glycine substitutions have reduced survival in contrast to those with small residue substitutions of glycine who tend to have a milder phenotype [26]. Furthermore, missense variants in the V-propeptide of the proα1(III)-chain are related to more evident clinical manifestations and vascular fragility. The substitutions of glutamic acid by lysine coexist and are related to dermatological manifestations that resemble the classical EDS combined with vascular and gastrointestinal disorders [27]. Finally, a few patients have been identified with biallelic COL3A1 variants who exhibit a severe phenotype with frequent polymicrogyria as a post-migration disorder [28].

As far as the clinical manifestations of the syndrome are concerned, patients with EDS experience a variety of symptoms, mainly concerning the GI tract, with the most common of them being nausea or vomiting, palpitations, abdominal pain and discomfort, and even inflammatory bowel disease. The severity of these manifestations depends on the type of the syndrome [29]. Τhe vEDS, also known as ‘the malignant type’, is associated with an increased incidence of GI disorders and serious complications. In particular, there is an increased risk of automatic bowel perforations, with many cases being reported in the literature [12]. The perforations mainly concern the colon, especially the sigmoid, followed by perforation of the final ileum and, much more rarely, of the stomach, caused basically by gastric volvulus, with a background of gastric ligaments abnormalities or diaphragmatic hernias combined with the rich blood supply of the stomach and its low ability to extend [30,31]. Perforations of the GI tract account for 82% of all the gastrointestinal complications in patients with vEDS, with a mean age of onset at 24 years old and a mortality rate of up to 12% [32]. In addition, in approximately 20% of the patients, the initial manifestation is any type of hernia formation, especially regarding vEDS [33]. Furthermore, rectal prolapse is another clinical complication of the syndrome, usually occurring in children or newborns and affecting a total of 4% of EDS patients, without any increased incidence observed for rectal prolapse with the vEDS [34]. However, the last one is associated almost exclusively with ruptures of the digestive tract, as already mentioned [29]. In addition, GI bleeding is the most common complication in patients with vEDS in the context of general vascular abnormality. It is reported that more than 50% of vascular complications involve the thoracic or the peritoneal cavity [32]. The medium-size vessels are usually involved, such as the mesenteric vessels and the renal or the splenic artery [35]. The pathophysiology behind the onset of gastrointestinal bleeding involves the formation of aneurysms, spontaneous ruptures of the vessel wall, or the formation of fistulas. Therefore, approximately 7% of patients present symptoms such as hematemesis, black stools, or intramural bleeding. Additionally, approximately 4% of the patients with EDS are diagnosed with gastrointestinal diverticulosis when undergoing an endoscopic examination [36].

Regarding our case, the patient did not have the characteristic facets described above; therefore, the diagnosis was not obvious. In fact, for months before the final diagnosis was established, the patient had either been hospitalized or undergone multiple surgical procedures due to complications involving the GI system in the form of intestinal perforations. In addition, he presented wound dehiscence, the formation of a postoperative abdominal hernia, and—even before the definitive diagnosis of the syndrome—a bleeding complication during the placement of a central venous catheter, where his vessels were appeared morphologically abnormal during the process of the port catheter placement by the vascular surgeons. Consequently, the patient presented the entire spectrum of complications of vEDS, although the physicians who first managed the patient were not suspicious of the syndrome.

The diagnosis of the syndrome is suspected through the complications which the patients usually present. However, the definite diagnosis is made exclusively by genetic testing and the detection of the pathological collagen genes. For the vascular type IV, a prenatal diagnosis is possible using polymorphic restriction genetic studies. In addition, the pathology reports in vEDS are not decisive for the diagnosis of the syndrome, as already mentioned. However, in our patient, segmental absence of the intestinal muscularis propria was observed.

Segmental absence of the intestinal musculature (SAIM) it is a rare clinical entity, characterized by the absence of the intestinal muscular layer. The disease is extremely rare and, as a result, the number of the cases being reported is very low, with less than 50 cases mentioned in the literature. The diagnosis of the syndrome is made only by histological confirmation of the absence of the intestinal muscular layer in the pathology reports. Patients present acute abdomen peritonitis due to the underlying perforation of the intestine. Treatment is surgical, with emergency exploratory laparotomy. SAIM is often associated with perinatal disorders and complications, and it is mostly diagnosed in neonates. Also, it is classified in two main subgroups: the idiopathic type and the secondary type of the disease [37].

As previously mentioned, in vEDS, there is no loss of intestinal muscle propria. The coexistence of these two isolated rare syndromes has been described only once, in a 16-year-old patient [38]. To the best of our knowledge, the index case is the second one in which the segmental absence of the intestinal muscularis and the vascular subtype of Ehlers–Danlos Syndrome coexist, increasing thereby the rarity of this case.

As far as treatment is concerned, there are no specific guidelines for the management of vEDS [8]. This lies in the rarity of the syndrome and in the fact that the diagnosis is made due to the occurrence of the complications, which are treated on an urgent basis. International data show that 10% of all the patients underwent surgery for a complication of the GI tract, while an increased incidence of 33% is reported in the vascular type [29]. In total, the need for surgery in patients with vEDS was about 68% and concerned arterial dissection, intestine, or other hollow viscus perforation. The total postoperative mortality at the point that it can be estimated remains low, about 2%. However, regarding only the complications of GI perforation or arterial complications, mortality appears increased by 45% and 41%, respectively [24]. The main impediment of these patients in the postoperative period is the generalized increased fragility of the tissues. Reperforations of the GI tract and ruptures or leaks of any intestinal or vascular anastomoses are quite common complications. Furthermore, all of the patients with EDS exhibit poor wound healing due to the lack of collagen types. As a result, tissue tearing from minimal handling, suture dehiscence, fistulas, and hernia formation are mutual complications with an occurrence of about 4–5% [39].

In our case, our patient underwent a total of seven surgical operations due to consecutive bowel perforations, while he also presented wound dehiscence and a postoperative abdominal hernia. In particular, these patients are 2.5 times more likely to develop an inguinal hernia than the rest of the population. Increased incidence occurs in young male patients, up to 19 years old, where the risk of hernia increases by about 10 times [40]. Moreover, patients with vEDS have an increased incidence of developing incisional hernias after abdominal operations due to excessively poor wound healing. Therefore, a mesh placement is considered indispensable for better postsurgical results [41]. However, it should be noted that, in the case of our patient, a mesh had been placed twice and, due to new bowel perforations, it was infected and was therefore finally removed. For all the reasons mentioned above, Hartmann’s procedure is a tested and relatively secure choice, especially for emergency surgery. In addition, some authors, in order to avoid complications of the ED disease, such as intestinal perforation or rupture of the anastomosis, suggest prophylactic total colectomy with ileorectal anastomosis, or even an end ileostomy, although the risk of rupture of ascendant, transverse, and descendant colon remains extremely low [1].

In summary, vEDS is an extremely rare genetic disorder. Patients present at an early age and the diagnosis is made through GI or vascular complications which are often the first manifestations. The most common perforation involves the sigmoid colon, which is the first clinical manifestation in the majority of patients, as shown in Table 1.

It should be mentioned that the strength of this study is that, in addition to the thorough case presentation and discussion, a systematic review of the literature was carried out, and all the vEDS patients who had a perforation of the GI tract as their first manifestation were collected. The rarity of the disease and the range of its clinical manifestations delay and further complicate its early diagnosis and appropriate therapeutic response. In our case, the diagnosis was made two years after the patient’s first episode of bowel perforation. Additionally, we describe for the second time, according to the existing literature, the coexistence of a vEDS patient with SAIM, increasing the rarity of this case presentation and, consequently, its significance.

Although there is no evidence so far that SAIM is caused by collagen pathology, it is noteworthy that this is the second report of SAIM and vEDS co-existence. The pathogenetic mechanism of SAIM is poorly understood, but the recurrent co-existence of the two entities raises the possibility of common etiology. If further investigation reveals that SAIM is part of the vEDS phenotype, this would contribute to the earlier diagnosis of vEDS in patients with atypical pathology findings, as in the index case.

The early diagnosis of vEDS is essential for better patient management, as well as improvements to their survival and quality of life. The diagnosis may also have implications for relatives, who are advised to undergo cascade testing.

However, this study has also some limitations that should be mentioned. Firstly, there was an inevitable information bias due to the fact that the patient was transferred to our surgical department from other hospitals. Therefore, all the information we have from before his admission to our department was collected from medical reports of previous hospitalizations in other trusts. Nevertheless, the information provided by other physicians is considered reliable enough in order to come to a safe conclusion based on it. Secondly, for the same reason, there is no intraoperative photographic material that could possibly contribute to a better presentation of the case.

## 4. Conclusions

The syndromes on the EDS spectrum constitute a challenge for physicians. Due not only to the specificity of the tissues, which are always fragile and difficult to manage surgically, but also the rarity of the syndrome, there are no specific guidelines for the management of complications such as enterocutaneous fistula and short bowel syndrome. In our case, the patient was admitted for the first time to the hospital for emergency treatment owing to intestinal perforation, without having any previous medical history. Based on the significance of an early diagnosis, it is important to raise awareness on the possibility of vEDS in young patients presenting spontaneous and recurrent perforations of the GI tract (even without specific findings on the pathology report).

## Figures and Tables

**Figure 1 jpm-13-01247-f001:**
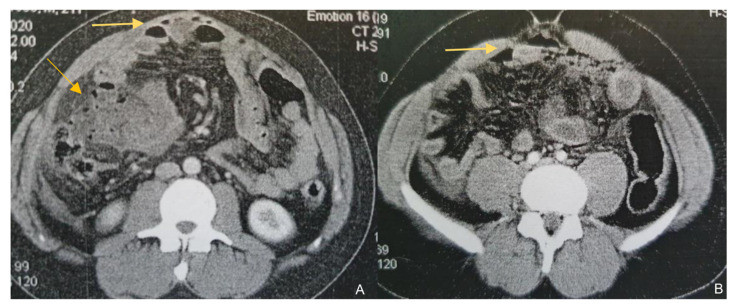
Contrast-enhanced CT findings. Axial images of contrast-enhanced CT on presentation (**A**) and a subsequent episode of perforation (**B**). In both cases, a note is made of marked fat stranding, free intraperitoneal gas bubbles, and small bowel wall thickening, suggestive of small bowel perforation.

**Figure 2 jpm-13-01247-f002:**
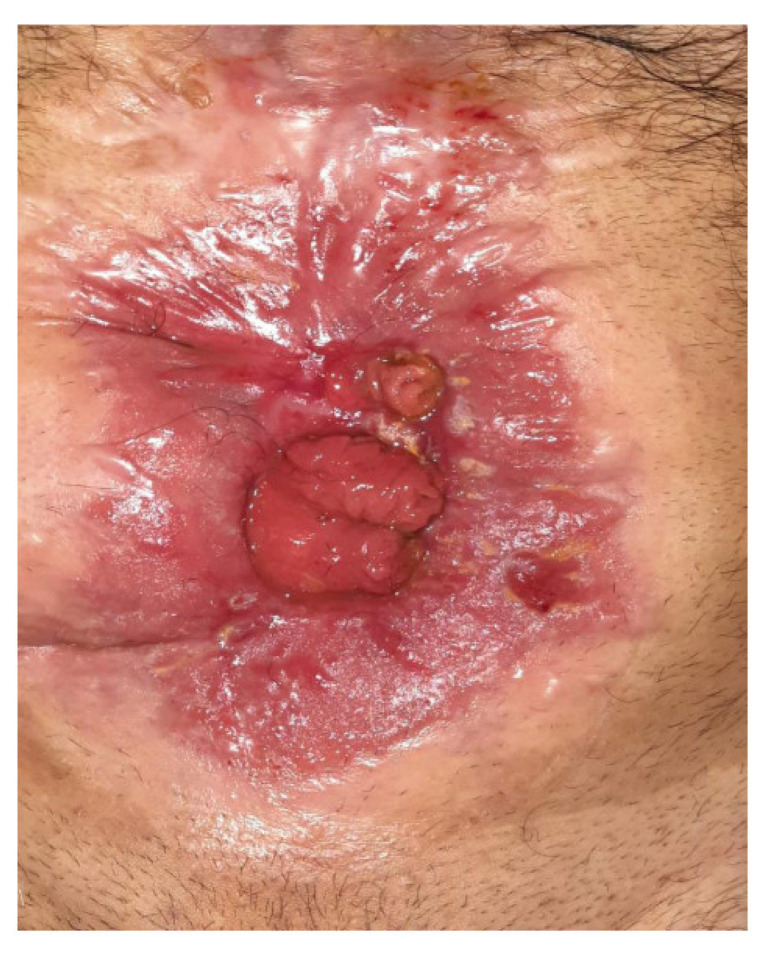
Postoperative Enterocutaneous Fistula.

**Figure 3 jpm-13-01247-f003:**
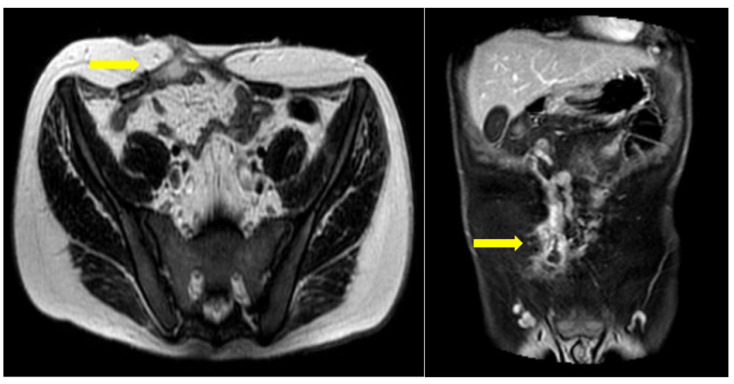
MRI enterography for follow-up. A gap in the anterior abdominal wall is seen, following open surgery with enterectomy. A peripherally enhancing fluid collection is seen juxtapositioned to the rectus abdominis muscle, in close connection with a collapsed ileal loop. This appearance is in keeping with a fistulous tract between adjacent bowel loops and the anterior wall.

**Figure 4 jpm-13-01247-f004:**
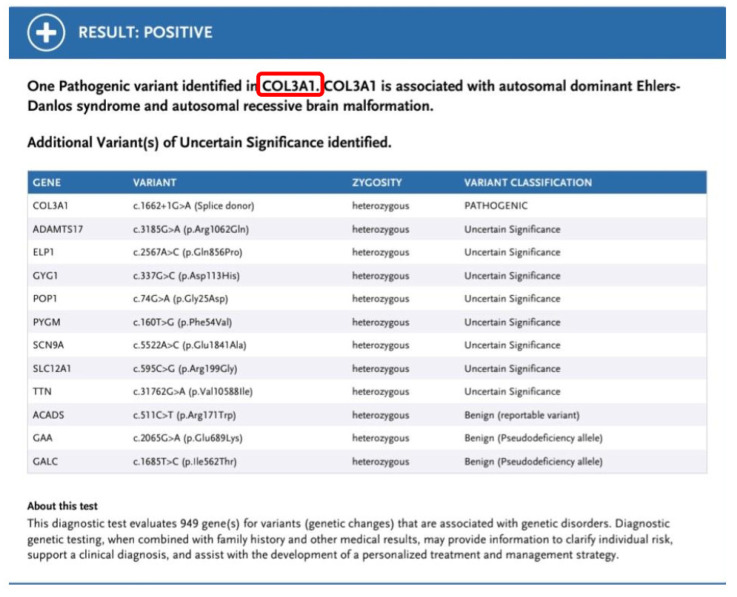
Genetic Results confirming the Syndrome.

**Table 1 jpm-13-01247-t001:** All the reported cases in the literature of patients with Vascular Ehlers-Danlos Syndrome manifesting bowel perforations.

	Reference	Journal/Year	Age/Gender	Location of Bowel Perforation	Management	Outcome
1	Edwin M. Sykes Jr.	1984	35 & 26 F	Sigmoid	End Colostomy/loop colostomy	-
2	Henry et al.	Arch Pediat, 1995	5/11&13 F	Sigmoid/colon twice	Colostomy/colectomy	-
3	Habib et al.	Ann R Coll Surg Engl, 2001	24 F	Right colon	Right hemicolectomy with a double-barrel stoma	
4	Solan et al.	Anesthesia, 2004	31 F	Transverse colon	Loop colostomy at the perforation site	
5	Sugawara	Surg Today, 2004	19 F	Sigmoid	Hartmann’s	
6	Fuchs & Fishman	J Pediatri Surg, 2004	14 F	Sigmoid	Loop colostomy	
7	Baichi et al.	Dig Dis Sci, 2005	48 F	Jejunum	Segmental resection	
8	Bedda et al.	Ann Chirurgie, 2006	23 M	Sigmoid	Loop colostomy	
9	Demirogullari et al.	J Pediatr Surg, 2006	15 F	Sigmoid	Hartmann’s	
10	Asherson et al.	J rhumatol, 2006	16 M	Sigmoid	Temporary defunction, ingcolostomy	
11	Blaker et al.	Virchows Arch, 2007	42 F	Sigmoid	Sigmoidectomy	Anastomotic leak, liver hematoma, death
12			21 M	Sigmoid	Hartmann’s	Good
13	Garvin et al.	J Trauma, 2008	27 F	Sigmoid	Primary repair	MVA 4 years later, small bowel multiple perforations, resection and end ileostomy
14	Guyot et al.	Br J Anaesth, 2009	7 M	Sigmoid	Laparotomy (-?)	Bleeding
15	Privitera et al.	Surg Today, 2009	35 F	Tranverse colon	Extended right hemicolectomy	Enterocutaneous fistula, splenic rupture, death
16	Leake et al.	Cases J, 2010	53 F	Distal ileum	Segmental resection	Good
17 18 19	Surgey et al.	Colorectal Dis	29 F	Sigmoid	Hartmann’s	good
			23 F	Sigmoid	Hartmann’s	-
			- F	Sigmoid	Hartmann’s	Good
20	Rana et al.	J Med Case Rep, 2011	33 M	Sigmoid	Hartmann’s	Good
21	Omori et al.	Surg Today, 2011	20 M	Sigmoid	Hartmann’s	Jejunal perforation, resection
22	Ng & Cheong	Asian J Surg, 2011	17 M	Sigmoid	Hartmann’s	-
23 24	Duthie et al.	J Ped Surg, 2011	12 F	Proximal rectum	Perforation oversewn	Good
			13 M	Colon	-	Colon reperforation twice
25	Allaparthi, et al	W J Gastrointest Endosc, 2013	23 F	Sigmoid	Colonic resection and ileostomy	Ileal perforation, bleeding
26	Shimaoka et al.	J Dermatol, 2013	11 F	Colon	-	Colon perforation twice
27	Kashizaki et al.	J Med Case Rep, 2013	64 F	-	Laparotomy & drainage	-
28	Eder et al.	Experimental Dermatol, 2013	23 F	Jejunum	Segmental resection	Good
29	Sa et al.	Surg Today, 2013	6 M	Sigmoid	Hartmann’s	Good
30	Anderson	Gastroenterology, 2014	48 F	Transverse colon	TAC & ileostomy	-
31	Yoneda et al.	Case rep Gastroenterol, 2014	20 F	Sigmoid	Hartmann’s	-
32	Inokuchi et al.	Medicine, 2014	24 M	Sigmoid	Hartmann’s	good
33	Nakagawa et al.	Intern Med, 2015	17 M	Colon	-	-
34	El Masri et al.	Techniques in Coloproctology, 2017	31 F	Sigmoid	Sigmoidectomy	reperforation
35	Our patient	2022	23 M	Sigmoid, multiple perforations of small intestine	Enterectomies	Enterocutaneous fistula, short intestine

## Data Availability

Informed consent was obtained from the patient.

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
