# Peer review of "Bowel Perforation in Vascular Ehlers–Danlos Syndrome: Case Report and Comprehensive Review"

_jpm, 2023, doi:10.3390/jpm13081247_

Round 1
Reviewer 1 Report
"Bowel Perforation in vascular EDS: a case report and a comprehensive review of the literature" is a well-studied and well-written manuscript. The case report is one-of-a-kind which highlights the lack of proper specific guideline for diagnosis/management of vEDS. The manuscript carries a lot of value. However, minor revisions are requested for betterment of the study.
1. Ethical concern: Please mention whether consent had been acquired from the patients for publishing the results and case history or not. If other ethical clearances are also required or have been acquired, it is recommended to properly mention them.
2. Figure 2 appears (line 64) before figure 1 (line 135).
3. Discussion is too lengthy with many repetitions. Discussion can be made concise or some part can be moved to introduction.
4. A table of summary of genes mutations that classify vEDS other than that has been mentioned in manuscript would benefit more to the readers.
5. Conservative therapeutic approach for vEDS can be explained a bit in the introduction.
Author Response
we would like to thank the reviewers for their valuable comments.

Reviewer 2 Report
The paper under review is well-written, accurate in its description, highly interesting, and could be very relevant to the scientific community. However, the entire paper lacks histological, immunohistochemical, and molecular analyses performed on the patient. A scientific paper requires accompanying evidence that thoroughly supports the claims made by the authors. Without this evidence, it is not possible to publish the paper, and it must be rejected.
For this paper to be published, the authors must include the results of the analyses that have been performed on the patient. Furthermore,
The authors should indicate whether the patient had any other symptoms, especially cutaneous manifestations, given the diagnosis that was subsequently made. Moreover, indicating the type of parenteral nutrition administered to the patient could be important to exclude any additional pro-inflammatory causes resulting from an inappropriate diet.
Line 21: "all" is repeated two times.
Figure 2: It would be relevant to highlight the interesting features concerning the figure, such as the detected pneumatosis, in order to make the results more accessible to an audience of non-medical healthcare professionals too.
Line 82: explain why the appendectomy was also performed.
Lines 90-95: Authors should include images of histological evaluations.
Line 110: Authors must include the immunohistochemical staining analyses
Line 127: Authors must include here a pathology report, possibly a representative histological analysis.
The English language is ok, minor revisions are required.
Author Response

(The authors gave the same response as above.)
